Application of bubble streams to control biofouling on marine infrastructure—pontoon-scale implementation

Hopkins Grant A. grant.hopkins@cawthron.org.nz
Scott Nicholas
Cahill Patrick
Cawthron Institute , Nelson , New Zealand
Waiho Khor
Electronic publication date: 2023 Sep 7
Publication date: 2023
Volume: 11
Electronic Location ID: e16004
Received 2023 Jun 22; Accepted 2023 Aug 9
Copyright: © 2023 Hopkins et al.
Copyright year: 2023
Copyright holder: Hopkins et al.
License: This is an open access article distributed under the terms of the Creative Commons Attribution License, which permits unrestricted use, distribution, reproduction and adaptation in any medium and for any purpose provided that it is properly attributed. For attribution, the original author(s), title, publication source (PeerJ) and either DOI or URL of the article must be cited.
License URL: https://creativecommons.org/licenses/by/4.0/

Keywords: Antifouling, Biosecurity, Biofouling management, Marina

Funding: New Zealand’s Ministry of Business, Innovation and Employment CAWX1904 The publication of this work was funded by New Zealand’s Ministry of Business, Innovation and Employment (CAWX1904–A toolbox to underpin and enable tomorrow’s marine biosecurity system). There was no additional funding received for this study. The funders had no role in study design, data collection and analysis, decision to publish, or preparation of the manuscript.

==============================
There is a lack of cost-effective, environmentally-friendly tools available to manage marine biofouling accumulation on static artificial structures such as drilling rigs, wind turbines, marine farms, and port and marina infrastructure. For there to be uptake and refinement of tools, emerging technologies need to be tested and proven at an operational scale. This study aimed to see whether biofouling accumulation could be suppressed on marine infrastructure under real-world conditions through the delivery of continuous bubble streams. Submerged surfaces of a floating marina pontoon were cleaned in-situ by divers, and the subsequent colonisation by biofouling organisms was monitored on treated (bubbles applied) and untreated sections. Continuous bubble streams proved highly effective (>95%) in controlling macrofouling accumulation on the underside surface of the marina pontoon for the first 2 months after deployment, but efficacy dropped off rapidly once bubble stream delivery was partially obscured due to biofouling accumulation on the diffuser itself. Although extensive macrofouling cover by mussels, bryozoans and hydroids was observed on treated surfaces by 4 months (27.5%, SE = 4.8%), biofouling % cover and diversity was significantly higher on untreated surfaces (79.6%, SE = 2.9%). While this study demonstrates that continuous bubble streams greatly restrict biofouling accumulation over short-to-medium timescales, improved system design, especially the incorporation of diffusers resistant to fouling, is needed for the approach to be considered a viable long-term option for biofouling management on static artificial structures.

Introduction

Accumulation of biological fouling (biofouling) on static artificial structures has cost and logistical implications for a broad range of maritime industries (see Hopkins et al. (2021a) for a review). Present-day management and mitigation actions range from periodic maintenance and reapplication of antifouling coatings (e.g., finfish predator nets) to installing structures without any protection and accepting the inevitable outcome (e.g., most port and marina infrastructure). Globally, marine biofouling management costs are enormous: for the aquaculture industry alone, they have been estimated at 5–10% of production costs, equating to US$1.5 to 3.5 billion (Fitridge et al. (2012) and references therein). In some settings, there are also indirect consequences for biofouling management inaction. For example, unmanaged biofouling in port and marina environments may spread onto nearby and distant habitats either via natural dispersal or human-mediated pathways (Forrest, Gardner & Taylor, 2009).

Proactive biofouling management of static marine infrastructure is hampered by the lack of cost-effective, environmentally-friendly tools. Mechanical methods are mostly used to remove biofouling once it has become well established rather than prevent it from accumulating in the first instance (Hopkins et al., 2021a). Emerging technologies or approaches, such as biological control (Switzer et al., 2011; Ross, Thorpe & Brand, 2004; Atalah et al., 2014), novel coatings and surface materials (Ware et al., 2018; Li & Guo, 2019; Wanka et al., 2020; Rawlinson et al., 2023) typically lack evidence of testing under real-world conditions at an operational scale, and this is slowing their uptake and further refinement (Hopkins et al., 2021a).

The aim of this study was to determine whether biofouling accumulation on a marina pontoon could be suppressed at scale under real-world conditions using continuous bubble streams. The present study builds on the strong foundation of previous studies that have attempted to control fouling on small-scale experimental surfaces (Scardino, Fletcher & Lewis, 2009; Bullard, Shumway & Davis, 2010; Lowen et al., 2016; Hopkins et al., 2021b) and sections of a stationary vessel (Scardino, Fletcher & Lewis, 2009). The premise of this approach is that continuous bubble streams physically remove (or ‘scour’) recently settled biofouling taxa (shear stress), and/or provide an impenetrable barrier between a vulnerable surface and larvae in the water column (Hopkins et al., 2021b). To test the approach at an operationally relevant scale, sections of a marina pontoon were subjected to treatment over 4 months. Efficacy was compared relative to corresponding control sections of the marina pontoon, yielding useful insights to guide future operational refinement of the system.

Materials and Methods

The study was conducted at the New Zealand Customs berth in Waikawa Marina, New Zealand (41°15′52.8″S, 174°02′17.5″E) over a period of high colonisation pressure (October 2022 to February 2023; late spring-late summer). To establish a ‘clean’ baseline, biofouling and biofilm were removed from the submerged surfaces of a marina pontoon by divers using handheld scrapers followed by pressure washing. Immediately following cleaning, two sections of the pontoon (approximately 10 m2 in total) were fitted with five 2-m long diffusers (pore size approximately 1 mm), connected using custom-made brackets (Fig. 1). The air diffusers were powered by a single air blower (K05 MS MOR 1.5 kW; FPZ Blower Technology, Concorezzo, Italy) via a series of hoses (40 mm internal diameter). Ball valves were fitted inline between the blower and diffusers so that flow rates could be independently adjusted to ensure bubble flow consistency across the treated areas.

Figure 1 Bracket schematics and CAD drawings of the pontoon with diffusers suspended beneath it.

Bubble streams were delivered continuously over a 4-month period. Diffusers were cleaned every 2 weeks to prevent macrofouling occlusion, with the exception of week 9 (originally scheduled for 21 December 2022), which was missed due to commercial diver availability. Efficacy of treatment was documented by obtaining high resolution photographs (Olympus TG6, 12-megapixel, fitted to a 210 × 320 mm quadrat) at the beginning of the trial (baseline), then every 4 weeks until completion. For each sampling event, 10 images were haphazardly taken within treated and untreated (control) sections; control sections were of the same dimensions as the treated sections, and were positioned immediately adjacent to, but outside the influence of, the bubble streams. At the completion of the trial, representative biofouling specimens were haphazardly sampled by divers and preserved in 70% ethanol to aid with subsequent image analyses. Photoquadrat images were analysed using the random dot method in Coral Point Count (CPCe V4.1; Kohler & Gill, 2006). One hundred stratified random points were overlaid on each image, and the area beneath each dot was categorised as either ‘bare space’, ‘biofilm’ or ‘macrofouling’. For macrofouling, taxa (>1 mm) were identified to major taxonomic groups.

All analyses were undertaken using R software (R Core Team, 2022). Generalised linear mixed models, with beta errors in the glmmTMB library (Brooks et al., 2017), were used to examine the effects of treatment and time on the percentage cover of bare space, biofilm, and macrofouling. Our models had treatment (two levels: Bubbled and Control) and months (four levels: months 0 to 4) as fixed orthogonal factors and block (two levels) as a random effect. Models were validated by inspecting simulated residuals using the Dharma R library (Hartig, 2022).

Results

Fouling cover and composition

Complete removal of macrofouling was achieved by divers scraping and water-blasting the submerged surfaces of the concrete pontoon, although a low cover of thin strips of biofilm remained (treatment averages = 5.0–9.2%; Figs. 2 and 3). During the first month, there was a comparable increase in biofilm cover within treated (average = 14.9%, SE = 2.6%) and untreated sections (average = 16.9%, SE = 2.1%), and after 2 months a red filamentous alga almost completely covered (94.7% cover, SE = 1.5%) the undersides of the untreated regions of the pontoon. By contrast, treated areas of the pontoon was sparsely covered by hydroids (average = 2.3%, SE = 0.8%) and biofilm (17.9%, SE = 3.0%).

Figure 2 Example images of biofouling accumulation on treated (Bubbled) and untreated (Control) sections of the marina pontoon through time.

Figure 3 Box plots of fouling percentage cover over time for treatment and control sections of the marina pontoon.

ns = not significant, **** = P < 0.0001.

An unanticipated lapse in bubble diffuser maintenance (week 9 of the experiment; see Methods) led to high levels of diffuser fouling, and a subsequent decline in treatment performance. A thick biofilm layer (average = 76.3%, SE = 4.7%) interspersed with a moderate coverage (23.3%, SE = 4.7%) of small black mussels (Xenostrobus pulex) was observed on treated surfaces, where previously only a biofilm was present (Fig. 3). Over this same period, there was a die-back in cover by the red filamentous alga on untreated surfaces, and the emergence of several mid-to-late succession fouling species (e.g., ascidians and bryozoans). After 4 months, at the completion of the experiment, treated surfaces had on average 27.5% macrofouling (mainly mussels, but also bryozoans and hydroids) and 72.5% (SE = 4.8%) thick biofilm coverage. Macrofouling coverage on untreated surfaces had increased significantly at this time point (p < 0.001; Table S1), averaging 79.6% cover (SE = 2.9%), and contained filamentous algae (58.5%), colonial ascidians (12.0%), tubeworms (5.1%), bryozoans (3.4%), solitary ascidians (<0.5%), mussels (<0.5%) and hydroids (<0.5%). Thus, despite the lapse in diffuser maintenance, macrofouling cover within treated areas was still significantly lower than that observed at the controls.

Discussion

Continuous bubble streams applied to the underside of a commercial floating pontoon proved effective in supressing marine biofouling for a period of 2 months, after which unmanaged diffuser fouling led to sub-optimal bubble delivery and the onset of biofouling cover on treated surfaces. Despite reactive diffuser maintenance after 3 months, the ongoing application of a bubble stream wasn’t sufficient to remove the fouling that had recently established in treated areas. By the completion of the experiment (4 months), macrofouling coverage and species composition in treated areas remained less abundant and diverse compared to controls, but still reached over 25% cover. Based on observations from previous field trials (Hopkins et al., 2021b), ongoing treatment was unlikely to remove existing fouling. Based on informal discussions with New Zealand marina pontoon manufacturers and marina managers, there is an expectation that biofouling treatments applied to marina pontoons should remain effective for at least 25–50% of their expected lifetime (ca. 50 years), with minimal ongoing interventions and maintenance costs. The current prototype system would not meet these criteria if applied at a marina-scale.

Our trial highlights two aspects of the prototype system that require improvement prior to further testing: (i) diffusers, hosing and brackets used to fix the protype are prone to becoming fouled; and (ii) shear forces produced by the bubble streams are insufficient to remove established biofouling (including soft foulers). There are emerging materials, coatings and surface treatments that may afford treatment-related equipment (including diffusers) biofouling protection for many months or even years (e.g., Lupoi et al., 2016; Rawlinson et al., 2023). However, based on a scan of existing and emerging products, it is unrealistic to expect a protection lifespan of >10 years for static infrastructure. Therefore, even with more advanced or less fouling-prone bubble delivery systems, periodic cleaning or replacement of diffusers would be required over the lifetime of a treated pontoon (or other permanent marine infrastructure). Given the shear forces required to remove established fouling, especially organisms that ‘cement’ themselves to a surface after initial colonisation (e.g., oysters and barnacles), intermittent spot cleaning of marina infrastructure will likely be required due to inevitable treatment failure at a range of spatial scales (e.g., irregularities in pontoon surfaces, unanticipated fouling of diffusers, power outages, downtime due to equipment damage). In anticipation of these scenarios, cost-effective methods to remove localised biofouling without the need for divers should also be explored (e.g., autonomous systems).

While not monitored as part of the trial, biofouling accumulation on the vertical sides of the pontoon treated by bubble streams had elevated levels of filamentous algae and mussels when compared with untreated portions (Fig. 4). To address this, future designs could position a diffuser at the edge of the pontoon so that the vertical surfaces are also subjected to treatment (see Hopkins et al. (2021b) for trials on vertical surfaces), or alternative approaches to fouling management could be implemented for the more accessible pontoon sides (e.g., biocontrol; Atalah et al., 2014).

Figure 4 High biomass of mussels (Xenostrobus pulex) and an unidentified filamentous alga was observed on the vertical surfaces of pontoons being treated with continuous bubble streams (A). Vertical surfaces adjacent to control areas had less biofouling biomass, and included bubble weed (Colpomenia sp.), bryozoans, colonial ascidians, hydroids, and small mussels (B).

Conclusions

Cost-effective marina pontoon antifouling systems are not commercially available. The prototype system deployed on a commercial marina pontoon in this study appears capable of keeping the underside surfaces free of biofouling, but only while the diffusers themselves remained free of biofouling. In the present configuration, this system is not viable for marina-scale applications, where effective timeframes of many years are expected. To meet these market expectations, improvements to the system (e.g., fouling resistant diffusers) or alternative approaches to deliver bubble streams should be investigated.

Supplemental Information

Supplemental Information 1 Results of generalised linear mixed models examining the fixed effects of treatment and months on the percentage cover of bare space, biofilm and macrofouling.

σ2 indicates the model residual variance. τ00 indicates the random effects variance component of Block. Significant values are indicated in bold. n = 197.

Click here for additional data file.

The authors thank Liam Falconer (Marlborough District Council), Anouk Euzeby and Gavin Beattie (Port Marlborough), James Brodie (Marlborough Commercial Diving Services), Javier Atalah (Cawthron Institute) for statistical advice, and anonymous reviewers for their logistical support and suggestions to improve the manuscript.

Additional Information and Declarations

Competing Interests

Author Contributions

Data Availability

The authors declare that they have no competing interests.

Grant A. Hopkins conceived and designed the experiments, performed the experiments, analyzed the data, prepared figures and/or tables, authored or reviewed drafts of the article, and approved the final draft.

Nicholas Scott conceived and designed the experiments, performed the experiments, prepared figures and/or tables, authored or reviewed drafts of the article, and approved the final draft.

Patrick Cahill conceived and designed the experiments, authored or reviewed drafts of the article, and approved the final draft.

The following information was supplied regarding data availability:

The data and R code is available at Zenodo: Javier Atalah. (2023). jatalah/bubble_streams_to_control_biofouling: v0.3 (v0.3). Zenodo. https://doi.org/10.5281/zenodo.8026013.

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
