# Peer review of "Application of bubble streams to control biofouling on marine infrastructure—pontoon-scale implementation"

_PeerJ, doi:10.7717/peerj.16004_

## Round 0.1 · original submission · Minor Revisions

I agree with the reviewers that this is a short yet concise article on the use of bubble streams for the control of biofouling, especially on ports and other marina structures. Please address all concerns raised by the reviewers and I look forward to reading the revised version of the manuscript.

·

Basic reporting

• Some minor formatting and style corrections are needed throughout, e.g.,
-- commas after e.g.
-- spaces between values and unit symbols.

Experimental design

• Methods
-- The authors have been forthcoming in identifying a key logistical issue (lapsed diffuser maintenance).
-- The methods and results lack some key pieces of information around the timing and duration of the maintenance lapse. See comments and suggested edits embedded in the attached document.

Validity of the findings

No comment.

Additional comments

• Introduction provides appropriate background, context and highlights key literature in the field.
• The research question is well-defined, and the authors highlight the knowledge gap the study attempts to address.
• The experimental setup is clearly presented and the quantitative methods are appropriate.
• The data are robust and statistically sound.
• The conclusions are well stated.

In the manuscript "Application of bubble streams to control biofouling on marine infrastructure - pontoon-scale implementation", the authors present an experiment that tests whether bubble streams have the potential to be an effective biofouling prevention tool at an operational scale.

The Introduction and the Discussion read well, but the Methods and Results lack some key pieces of information around the timing and duration of the maintenance lapse (see specific comments in the manuscript). This maintenance lapse is an important part of the narrative, as it demonstrates a key limitation that needs to be overcome: diffusers themselves are prone to biofouling, which in turn limits the efficacy of this approach.

The findings provide insights towards operational considerations and obstacles for developing bubble curtain systems that are effective for proactive biofouling management. This contribution will be of interest to researchers, developers of biofouling prevention technologies, and marine infrastructure managers.

-- Daniel Kluza

·

Basic reporting

The manuscript was brief but well written, with clear and concise rationale for it's need and for the results observed. There was an appropriate number and scope of references, with a solid understanding of the background problem being addressed and the previous work undertaken in this area.
No issues.

Experimental design

The experimental design was appropriate and well designed for the study at hand. Given the scale of the equipment and resources required, the experiment had sufficient replication, with statistical approaches (e.g. blocking) used to bolster the strength of the results.
No issues.

Validity of the findings

While not entirely novel in its subject matter, the study does a solid job of investigating the utility of bubbled air for biofouling control on port/marina infrastructure. The analysis of the data is appropriate and robust for the questions/study at hand. The conclusions are brief but concise, and clearly state the outcome of the work. It is encouraging to see that the authors acknowledge and discuss the "real world" shortcomings of their technology/approach and how the delivery and/or technology must be improved in order to achieved the desired benefits and outcome. This later point should be encouraged.

Additional comments

The manuscript was brief but well written, with clear and concise rationale for it's need and for the results observed. There was an appropriate number and scope of references, with a solid understanding of the background problem being addressed and the previous work undertaken in this area. The study does a solid job of investigating the utility of bubbled air for biofouling control on port/marina infrastructure. The analysis of the data is appropriate and robust for the questions/study at hand. The conclusions are brief but concise, and clearly state the outcome of the work. It is encouraging to see that the authors acknowledge and discuss the "real world" shortcomings of their technology/approach and how the delivery and/or technology must be improved in order to achieved the desired benefits and outcome. This later point should be encouraged.

Reviewer 3 ·

Basic reporting

no comments

Experimental design

no comment

Validity of the findings

no comment

Additional comments

no comment

---

## Round 0.2 · accepted · Accept

Albeit short, the manuscript presents some novel insights into the use of bubble streams to control biofouling. The current version of the manuscript is acceptable for publication.